# Susceptibility to Pentylenetetrazole-Induced Seizures in Mice with Distinct Activity of the Endogenous Opioid System

**DOI:** 10.3390/ijms25136978

**Published:** 2024-06-26

**Authors:** Anna Ruszczak, Piotr Poznański, Anna Leśniak, Marzena Łazarczyk, Dominik Skiba, Agata Nawrocka, Kinga Gaweł, Justyna Paszkiewicz, Michel-Edwar Mickael, Mariusz Sacharczuk

**Affiliations:** 1Department of Small Animal Diseases with Clinic, Faculty of Veterinary Medicine, Warsaw University of Life Sciences, Nowoursynowska 166, 02-787 Warsaw, Poland; 2Department of Experimental Genomics, Institute of Genetics and Animal Biotechnology, Polish Academy of Sciences, Postępu 36A, 05-552 Jastrzębiec, Poland; 3Laboratory of Host-Microbiota Interactions, Nencki Institute of Experimental Biology, Polish Academy of Sciences, Pasteura 3, 02-093 Warsaw, Poland; 4Department of Pharmacotherapy and Pharmaceutical Care, Faculty of Pharmacy, Medical University of Warsaw, Banacha 1, 02-697 Warsaw, Poland; 5Department of Experimental and Clinical Pharmacology, Medical University of Lublin, Jaczewskiego 8b, 20-090 Lublin, Poland; 6Department of Health, John Paul II University of Applied Sciences in Biala Podlaska, Sidorska 95/97, 21-500 Biała Podlaska, Poland

**Keywords:** seizures, epilepsy, endogenous opioid system

## Abstract

Currently, pharmacotherapy provides successful seizure control in around 70% of patients with epilepsy; however, around 30% of cases are still resistant to available treatment. Therefore, effective anti-epileptic therapy still remains a challenge. In our study, we utilized two mouse lines selected for low (LA) and high (HA) endogenous opioid system activity to investigate the relationship between down- or upregulation of the opioid system and susceptibility to seizures. Pentylenetetrazole (PTZ) is a compound commonly used for kindling of generalized tonic-clonic convulsions in animal models. Our experiments revealed that in the LA mice, PTZ produced seizures of greater intensity and shorter latency than in HA mice. This observation suggests that proper opioid system tone is crucial for preventing the onset of generalized tonic-clonic seizures. Moreover, a combination of an opioid receptor antagonist—naloxone—and a GABA receptor agonist—diazepam (DZP)—facilitates a significant DZP-sparing effect. This is particularly important for the pharmacotherapy of neurological patients, since benzodiazepines display high addiction risk. In conclusion, our study shows a meaningful, protective role of the endogenous opioid system in the prevention of epileptic seizures and that disturbances in that balance may facilitate seizure occurrence.

## 1. Introduction

Epilepsy is the most common neurological condition, with gaps in knowledge about its etiopathology. About 1% of the human population suffers from epilepsy, with almost one-third having the refractory form of the disease [1]. Therefore, there is still a justified need to seek novel antiseizure medications (ASMs) [2]. Every year, nearly 50 new epilepsy cases per 100,000 inhabitants worldwide are recognized [3]. According to the World Health Organization (WHO), the prevalence of epilepsy is 40–70 new diagnoses per 100,000 in developed countries and twice as much in developing regions of the world. About 75% of epilepsy cases are diagnosed in childhood, which reflects the increased susceptibility of the developing brain to seizures resulting from traumas or infectious diseases, which occur more often at younger ages [2]. Apart from very rare cases of epileptic seizures being a result of developmental malformations (e.g., ectopic cerebellum arising outside the posterior fossa), most epilepsy cases can be controlled, or even prevented, by reducing exposure to known risk factors [4]. The clinical definition of epilepsy describes seizures as a symptom, not a disease per se [5]. Typically, seizures are a consequence of disruptions in the fragile balance between neural excitation and inhibition in the brain, maintained by a plethora of neurotransmitters [6]. One well-recognized hypothesis explaining the molecular underpinnings of seizure onset is neurotransmitter imbalance following a massive, uncontrolled release of glutamate paired with insufficient γ-aminobutyric acid (GABA)-mediated inhibitory neurotransmission [7,8,9]. Currently, epilepsy cases of formerly unknown origin are classified as having a strong genetic background [10]. ASMs utilized in present clinical practice fail to control seizures in all cases.

Endogenous mechanisms ensuring optimal neurotransmitter tone are one of the natural strategies for seizure prevention. There are multiple endogenous neuropeptides such as neuropeptide Y, galanin, somatostatin, ghrelin and opioid peptides that modulate neural excitability in the central nervous system (CNS). These molecules exhibit direct anticonvulsant properties and are perceived as a good starting point for the design of new promising ASMs [11]. Studies on patients with focal epilepsy have documented that seizures may arise from the dysregulation of opioid receptor-mediated G-protein signaling and the PKA/CREB and DAG/IP3 effector pathways [12]. An increase in opioid receptor availability was also demonstrated in temporal lobe seizure patients [13]. Another study has confirmed that the endogenous opioid system plays an important neuroprotective role in the CNS [14], which may prevent seizures, reduce their frequency and intensity or facilitate recovery after their occurrence. Local kainic acid-induced epileptic events in the hippocampus may be overcome through the activation of κ opioid receptors that suppress neuronal excitation, contributing to a decrease in seizure intensity and reducing consequent neurodegeneration [15]. A study by Loacker and colleagues implies that prodynorphins that act as endogenous κ opioid receptor agonists have the most meaningful impact on the modulation and control of excitatory neurotransmission in hippocampal granule cells during epileptogenesis and thus may serve as potential treatment targets [16]. Other preclinical investigations also confirm that various selective κ opioid receptor agonists produce time- and dose-dependent anticonvulsive effects [16,17]. This observation was also validated in a clinical trial, where patients with decreased prodynorphin levels were more susceptible to seizures [8]. It is widely acknowledged that κ opioid receptors, apart from producing dysphoria, are protective against seizures. On the other hand, activation of the μ opioid receptors by exogenous ligands was shown not only to generate a euphoric effect but also to induce convulsions [8,18]. Studies on multiple animal models have confirmed that selective blockage of μ opioid receptors inhibits seizure-like activity [19,20,21,22]. Considering δ opioid receptors, proconvulsive effects are most commonly reported [23,24,25]. However, δ opioid receptor activation has neuroprotective effects as well, as determined in hypoxic and ischemic models [26].

In our study, we utilized the most common and reproducible chemically-induced temporal lobe seizure kindling model involving repeated low-dose pentylenetetrazole (PTZ) injections [1,27,28]. To determine the participation of the opioid system in the development of seizures, we used a model of Swiss Webster mice selectively bred towards high (HA) and low (LA) analgesia induced by swim stress. The HA and LA mouse lines display molecular backgrounds rendering their endogenous opioid system either up- or downregulated, respectively. Therefore, this model is well-suited for seeking new mechanisms linked with the opioid system in multiple CNS pathologies, such as neurological or neuropsychiatric disorders. Due to the lack of effective therapies for refractory epilepsy, there is still a necessity to explore new mechanisms of the disease and define new drug targets, bearing in mind that they may be related to the congenital divergence in opioid system activity [29,30].

### Current Pharmacotherapy of Epilepsy and Its Relationship with Opioid Signaling

The most common ASMs used currently in clinical practice, included in the recommendations of the International League Against Epilepsy and National Institute for Health and Care Excellence (NICE) guidelines, for the treatment of generalized or focal seizures, are sodium valproate, lamotrigine and levetiracetam. Some other drugs are well-known and almost historical, having been applied as anticonvulsants for decades, but recommended with caution if the drugs of the first-line, or even second- or third-line therapy are found to be ineffective, mostly because of the risks of serious side effects and the availability of the concurrent, relatively newly-developed, pharmacological interventions with better safety profiles. Among the “older” medications are phenytoin, carbamazepine, gabapentin, pregabalin, clonazepam and several more.

Sodium valproate has been shown to increase inhibitory GABA neurotransmitters in the CNS. When the drug is applied in combination with morphine, it prevents the development of a tolerance to the opioid, as well as decreases activation of glycogen synthase kinase-3 β (GSK3β) in mice brains [31]. Lamotrigine is an anticonvulsant medication that blocks voltage-gated sodium channels. Acting presynaptically, it inhibits excitatory glutamate release, therefore activation of N-methyl-D-aspartate (NMDA) and non-NMDA receptors, and consequently reduces the NO synthesis that accompanies opiate abstinence. Lamotrigine, at a dose of 100 mg/kg, administered 45 min before naloxone (1 mg/kg) injection for withdrawal symptom-induction in mice treated for 4 days with morphine (75 mg implanted pellet) almost completely eliminated the abstinence manifested by micturition, diarrhea, sniffing and grooming, tremor and shaking [32]. Levetiracetam’s mechanism of action relies on the modulation of synaptic neurotransmitter release by binding to the synaptic vesicle protein SV2A. GABAergic terminals in some brain regions co-express this membrane transporter; however, glutamatergic endings of other cerebral areas also demonstrate SV2A’s presence, hence its role in enhancing or impairing neural transmission [33]. Levetiracetam has been demonstrated to reduce inflammatory hyperalgesia in rats, which is partly mediated by GABA_A_, opioid, 5-HT and α2-adrenergic receptors. Since levetiracetam selectively inhibits the N-type Ca^2+^ channels and reduces Ca^2+^ conductance, similarly to the way opioid receptor agonists do, levetiracetam may contribute to opioid-induced antinociception [34]. Another anticonvulsant drug, phenytoin, is known to stabilize the inactive state of the voltage-gated sodium channels and was documented to improve the antinociceptive effect of morphine in a neuropathic pain model [35]. Similarly, carbamazepine potentiates morphine efficacy in rats with neuropathic pain [36]. The drug binds to the inactivated Na^+^ channels and decelerates the regeneration of inactivation. It also delays calcium ion entry into synaptic membranes. Finally, carbamazepine may impede catecholamine uptake and increase GABA, but it decreases glutamate levels, reducing neuronal excitability. Gabapentin displays an affinity for the binding sites of the voltage-gated calcium channels, which contributes to the inhibition of excitatory amino acids released from presynaptic areas. The drug is used also for the treatment of neuralgia caused by nerve damage and peripheral diabetic neuropathy. Gabapentin is also known to potentiate the morphine-induced analgesia effect of morphine [37]. Pregabalin is structurally similar to gabapentin. The drug interacts with voltage-gated calcium channels and demonstrates a comparative antipain effect to tramadol in an acute mouse model of pain [38]. Clonazepam, a long-acting benzodiazepine that behaves as a GABA_A_ receptor agonist, may also promote serotonin synthesis. Interaction with opioids may be fatal if overdosed, e.g., with oxycodone [39]. To summarize, combined treatments consisting of ASMs that reduce neuronal excitation and opioids, acting via receptors coupled with inhibitory G_i_ proteins, may enhance healing effects but also pose a risk for drowsiness or even respiratory depression if used inappropriately.

## 2. Results

### 2.1. Susceptibility of HA and LA Mice to the Proconvulsive Activity of PTZ (Acute Administration)

Pentyenetetrazole (PTZ) injected intraperitoneally (ip.) dose-dependently induced seizures in mice of both lines [F(2,54) = 14.77, *p* < 0.001—intensity; = 14.34, *p* < 0.001—latency], with higher intensity and shorter latency in LA mice [F(1,54) = 34.82, *p* < 0.001—intensity; = 51.85, *p* < 0.001—latency]. The dose-dependent effect of PTZ for both analyzed parameters was more robust in the HA [F(2,27) = 10.14, *p* < 0.001—intensity; = 9.76, *p* < 0.001—latency] than in the LA line [F(2,27) = 4.77, *p* < 0.05—intensity; = 7.72, *p* < 0.01—latency], predominantly due to the lack of PTZ efficacy at the lowest dose in HA mice (Figure 1 and Figure 2).

### 2.2. Effect of Opioid Receptor Blockage on Acute PTZ-Induced Seizures in HA and LA Mice

Naloxone (NLX) enhanced the proconvulsive activity of PTZ only in HA mice by increasing seizure intensity and decreasing seizure latency (Figure 3 and Figure 4), as confirmed by the significant interaction between line and treatment [F(1,114) = 23.26, *p* < 0.001—intensity; F(1,114) = 25.86, *p* < 0.001—latency]. In the HA line, the NLX-induced increase in seizure intensity and decrease in seizure latency were independent of the PTZ dose. Therefore, the effect of the interaction between line, naloxone and PTZ was found to be negligible [F(2,114) = 1.27, *p* = 0.284—intensity; F(2,114) = 2.77, *p* = 0.066—latency]. Selective blockage of µ opioid receptors by β-FNA diminished seizure-like behavior in HA mice. The anticonvulsive effect of β-FNA was confirmed by a significant drug and line interaction [F(1,115) = 4.28, *p* < 0.05—intensity; F(1,115) = 4.17, *p* < 0.05—latency], but only when seizures were induced with 100 mg/kg of PTZ (Figure 3 and Figure 4) (line x β-FNA × PTZ interaction: F(2,115) = 0.47, *p* = 0.624—intensity; F(2,115) = 0.32, *p* = 0.727—latency]. Administration of β-FNA had no effect on seizure intensity and seizure latency in the LA line. All fixed doses of β-FNA had a similar influence on HA mice when compared to the effects of interactions considering line type, dose of PTZ and µ opioid receptor (MOR) antagonist [F(2,115) = 0.47, *p* = 0.624—intensity; = 0.32, *p* = 0.727—latency].

Conversely, selective blockage of the δ and κ opioid receptors by NTI [F(1,122) = 50.86, *p* < 0.001—intensity; F(1,122) = 49.89, *p* < 0.001—latency] and nor-BNI [F(1,114) = 15.53, *p* < 0.001—intensity; F(1,114) = 32.41, *p* < 0.001—latency], respectively, resulted in increased seizure intensity and lower seizure latency in HA but not in LA mice (Figure 3 and Figure 4). The effect of δ opioid receptor blockage by NTI on seizure intensity and seizure latency was extremely robust even at the lowest dose of PTZ used [F(2,122) = 4.61, *p* < 0.05—intensity; F(2,122) = 5.22, *p* < 0.01—latency]. In HA mice, nor-BNI had a similar potentiating effect on seizure-like behavior when 50 or 75 mg/kg of PTZ was used [F(2,114) = 1.42, *p* = 0.247—intensity; (2,114) = 2.61, *p* = 0.078—latency].

### 2.3. Effect of Opioid Receptor Agonists and Diazepam on Acute PTZ-Induced Seizures in HA and LA Mice

Pre-treatment of HA or LA mice with morphine (Morph) had no significant modifying effect on seizure intensity or seizure latency in response to PTZ [F(1,116) = 0.14, *p* = 0.714—intensity; F(1,116) = 0.01, *p* = 0.960—latency] (Figure 5 and Figure 6). This was confirmed by the lack of interaction between line and morphine treatment [F(1,116) = 2.79, *p* = 0.098—intensity; F(1,116) = 0.41, *p* = 0.523—latency]. Morphine was also ineffective regardless of the PTZ dose used (line × morphine × PTZ interaction: F(1,116) = 0.46, *p* = 0.630—intensity; F(1,116) = 0.09, *p* = 0.917—latency]. On the other hand, both buprenorphine and biphalin increased seizure intensity in HA mice (Figure 5 and Figure 6), as confirmed by the significant line × treatment interaction [buprenorphine: F(1,114) = 20.24, *p* < 0.001—intensity; F(1,114) = 26.73, *p* < 0.001—latency], [biphalin: F(1,112) = 6.49, *p* < 0.05—intensity; F(1,112) = 15.05, *p* < 0.001—latency]. Both buprenorphine and biphalin intensified seizures and shortened seizure latency in HA mice independent of the PTZ dose used, as three-way ANOVA failed to detect a significant line × treatment × PTZ interaction [buprenorphine: F(2,114) = 0.76, *p* = 0.471—intensity; F(2,114) = 0.68, *p* = 0.484—latency], [biphalin: F(2,112) = 0.35, *p* = 0.707—intensity; F(2,112) = 0.64, *p* = 0.530—latency].

Administration of diazepam (DZP) strongly inhibited the proconvulsive activity of PTZ [F(1,114) = 349.27, *p* < 0.001—intensity; F(1,114) = 175.60, *p* < 0.001—latency] in both mouse lines at each dose of PTZ used (Figure 5 and Figure 6). Because of lower seizure latency, the anticonvulsant activity of diazepam in PTZ-treated LA mice was more pronounced than in HA mice [line × DZP interaction: F(1,114) = 46.24, *p* < 0.001—intensity; F(1,114) = 37.01, *p* < 0.001—latency]. The effect of diazepam was particularly prominent when the lowest dose of PTZ was used, which induced seizures only in LA mice [line × diazepam × PTZ dose interaction: F(2,114) = 6.81, *p* < 0.01—intensity; F(2,114) = 4.69, *p* < 0.05—latency].

### 2.4. Modulation of the Anticonvulsive Effect of Diazepam by Naloxone in HA and LA Mice Acutely Treated with PTZ

Administration of DZP (1 and 5 mg/kg) prior to PTZ, decreased both seizure intensity and latency [F(1,152) = 225.14, *p* < 0.001—intensity; F(1,152) = 62.70, *p* < 0.001—latency] in both LA and HA mice (Figure 7 and Figure 8), as confirmed by the lack of significant interaction between line and DZP treatment [F(1,152) = 0.24, *p* = 0.623—intensity; F(1,152) = 0.01, *p* = 0.961—latency]. As expected, the higher dose of diazepam (5 mg/kg) produced a stronger anticonvulsive effect [F(2,148) = 634.51, *p* < 0.001—intensity; = 112.24, *p* < 0.001—latency], particularly in HA mice, as shown by a significant line × diazepam interaction [F(2,148) = 4.02, *p* < 0.05—intensity; = 5.74, *p* < 0.01—latency]. Administration of NLX, markedly enhanced the anticonvulsant activity of diazepam, as confirmed by the NLX × DZP interaction [F(1,152) = 51.19, *p* < 0.001—intensity; F(1,152) = 38.88, *p* < 0.001—latency]. Interestingly, this effect was independent of the mouse line, as shown by the non-significant line × NLX × DZP interaction: [F(1,152) = 0.77, *p* = 0.380—intensity; = 3.27, *p* = 0.073—latency]. Importantly, diazepam (1 mg/kg) and NLX co-treatment produced a similar anticonvulsive effect to 5 mg/kg of diazepam alone. Moreover, the effect of NTX, depending on the dose of DZP, was stronger at lower doses in mice of the HA line [interaction effect line × NTX × DZP dose: F(2,148) = 5.58, *p* < 0.01—intensity; = 47.85, *p* < 0.001—latency].

### 2.5. Susceptibility of HA and LA Mice to PTZ-Kindled Seizures

Chronic administration of a subliminal dose (20 mg/kg) of PTZ triggered seizures in both HA and LA mice when challenged with a last single bolus of 50, 75 and 100 mg/kg PTZ (Appendix A). Both seizure intensity and seizure latency following chronic PTZ administration were dose-dependent [F(2,64) = 33.58, *p* < 0.001—intensity; = 14.96, *p* < 0.001—latency]. LA mice experienced greater seizure intensity than their HA counterparts [F(1,64) = 4.09, *p* < 0.05]. However, both lines did not differ in terms of seizure latency, as evidenced by an insignificant effect of line [F(1,64) = 2.83, *p* = 0.097]. The significant effect of the last dose of PTZ on the level of seizure intensity or latency was revealed by one-way ANOVA in HA mice [mean effect of PTZ dose: F(2,33) = 22.24, *p* < 0.001—intensity; = 9.99, *p* < 0.001—latency] and within LA line [mean effect of the PTZ dose: F(2,31) = 11.77, *p* < 0.001—intensity; = 5.13, *p* < 0.05—latency]. Chronic injections of PTZ enhanced seizure intensity and shortened latency as compared to relevant groups of mice receiving only a single dose of PTZ (Appendix A) [mean effect of type of procedure: F(1,124) = 6.86, *p* < 0.01—intensity; = 6.61, *p* < 0.05—latency] and this observation was specific only to HA mice, confirmed following an analysis of mouse line–procedure interaction [F(1,124) = 15.01, *p* < 0.001—intensity; = 15.90, *p* < 0.001—latency]. Furthermore, an increase in the seizure intensity of HA mice subjected to the chronic procedure was noted, with any last PTZ dose found to be effective [effect of interaction among mouse line, type of procedure and PTZ dose: [F(2,124) = 0.70, *p* = 0.498—intensity; = 0.53, *p* = 0.587—latency].

#### 2.5.1. The Effect of Chronic Naloxone Treatment in HA and LA Mice Kindled with PTZ

Multiple injections of NLX did not affect seizure intensity or seizure latency in mice of either line kindled with PTZ (Appendix A). This was confirmed by an insignificant main effect of naloxone [F(1,130) = 0.75, *p* = 0.387—intensity; F(1,130) = 0.70, *p* = 0.403—latency] followed by a negative NLX × line interaction [F(1,130) = 0.01, *p* = 0.923—intensity; F(1,130) = 0.01, *p* < 0.943—latency]. Naloxone was ineffective at any PTZ dose used [NLX × PTZ interaction: F(2,130) = 0.30, *p* = 0.744—intensity; F(2,130) = 0.26, *p* = 0.775—latency], with neither having a line-specific effect [NLX × PTZ × line interaction: F(2,130) = 0.25, *p* = 0.779—intensity; F(2,130) = 0.81, *p* = 0.449—latency]. However, when seizure-like behavior in HA mice treated acutely with PTZ was compared with the PTZ-kindled model, the proconvulsive effect of NLX was less robust in PTZ-kindled HA mice both in terms of seizure intensity and seizure latency. This conclusion was drawn based on the significant main effect of model type [F(1,260) = 9.68, *p* < 0.01—intensity; F(1,260) = 7.66, *p* < 0.01—latency] as well as a significant interaction between line, model and NLX [F(1,260) = 4.10, *p* < 0.05—intensity; F(1,260) = 4.03, *p* < 0.05—latency].

#### 2.5.2. Effect of Chronic Morphine Treatment on the Proconvulsive Effect of Repeated PTZ Administration in HA and LA Mice

As shown in Appendix A, chronic morphine treatment did not affect PTZ-induced seizure intensity or seizure latency in either mouse line, as evidenced by the insignificant main effect of morphine [F(1,114) = 0.08, *p* = 0.781—intensity; F(1,114) = 0.10, *p* = 0.750—latency] and an insignificant line × Morph interaction [F(1,114) = 0.23, *p* = 0.631—intensity; F(1,114) = 0.05, *p* < 0.823—latency]. However, a significant Morph × PTZ interaction suggested that morphine might exert a dualistic effect dependent on the dose of PTZ used [F(2,114) = 8.50, *p* < 0.001—intensity; F(2,114) = 7.41, *p* < 0.001—latency]. Namely, when the lowest dose of PTZ was used (50 mg/kg), morphine lowered seizure intensity and increased seizure latency (*p* < 0.05). Surprisingly, at 75 mg/kg of PTZ, morphine produced an opposite effect by intensifying seizures and shortening seizure latency (*p* < 0.05) [Morph × PTZ interaction F(2,58) = 7.47, *p* < 0.01—intensity; F(2,58) = 6.23, *p* < 0.01—latency]. This dualistic pattern of morphine’s effect was not seen in LA mice, as confirmed by a non-significant line × Morph × PTZ interaction [F(2,114) = 0.51, *p* = 0.602—intensity; = 0.58, *p* = 0.559—latency]. This dualistic pattern of morphine’s effect was not seen in LA mice, as confirmed by a non-significant line × Morph × PTZ interaction [F(2,56) = 2.15, *p* = 0.126—intensity; = 1.87, *p* = 0.163—latency].

Interestingly, the impact of morphine on seizure latency relied on the type of seizure model used (acute vs. kindled). This was evidenced by a significant Morph × PTZ × model interaction: [F(2,115) = 3.18, *p* < 0.05]. However, this effect was not seen in relation to seizure intensity [F(2,115) = 1.95, *p* = 0.147]. Similar observations were not found in LA mice, as confirmed by a non-significant interaction between morphine, PTZ and model type: F(2,115) = 1.79, *p* = 0.172—intensity; F(2,115) = 0.88, *p* = 0.418—latency].

## 3. Discussion

Opioids are endogenous peptides that modulate the excitability of neurons in the CNS and display pro- or anticonvulsive effects, depending on the type of opioid ligand, its dose, route of administration or type of targeted opioid receptors [23,40,41,42,43,44,45].

In the present study, by employing a mouse model of high and low endogenous opioid system activity, we have broadened the knowledge of the opioid system’s involvement in the modulation of seizures. The obtained results suggest that the downregulation of endogenous opioid system activity in LA mice decreases seizure latency and increases seizure severity upon acute pentylenetetrazole (PTZ) administration as compared to HA mice. The two mouse lines utilized in the present study demonstrate several key features that may explain their varied responses to chemically-induced seizures. HA mice, compared to their LA counterparts, display elevated β-endorphin levels in the brain and significantly greater G protein activation in opioid receptor-rich regions like the thalamus, hypothalamus, periaqueductal grey matter and hippocampus [46]. HA mice also show increased permeability of the blood-brain barrier due to lower expression levels of occludin and claudin-5, which may facilitate neurotransmitter leakage (including endogenous opioid peptides) and may trigger their compensatory overexpression [47].

In our study, opioid system blockage by naloxone (10 mg/kg) increased the proconvulsive effects of acute PTZ administration in HA mice in a dose-dependent manner. Similar proconvulsive effects following naloxone administration were reported in previous studies on rats [48], mice [49] and primates [50]. Opioid receptors, as a member of the G protein-coupled receptors family, interact with inhibitory G_i/o_ proteins. Upon agonist binding, opioid receptors couple to G proteins, which leads to a downstream signal cascade. Stimulation of opioid receptors of µ and δ type by an agonist inhibits adenyl cyclase and voltage-gated Ca^2+^ channels, as well as induces the opening of K^+^ channels, resulting in a decrease in neuronal excitability and reduction in neurotransmitter release [51]. In general, opioid-mediated attenuation of neural transmission is a result of a reduction of neurotransmitter release from presynaptic areas and a hyperpolarisation of postsynaptic neurons [40,51,52]. This evidence on opioid systems suggests that its blockade by naloxone may result in the inverse effect; however, the final outcome of the opioid antagonist treatment is dependent on many more factors, other endogenous systems’ activity, levels of distinct neurotransmitters and mediators and the dynamic balance between them. Interestingly, several early reports suggested that the seizure-promoting effects of naloxone are at least partly opioid receptor-independent and this compound was found to act as a negative modulator of gamma-aminobutyric acid (GABA) receptors [53,54]. Considering LA mice, the observed ineffectiveness of naloxone in LA mice with low activity of the endogenous opioid system may be explained by significantly affected G protein activity in these animals’ brains, as compared to their HA counterparts [46].

On the contrary, in the PTZ-kindling model of seizures, chronic naloxone was ineffective in modulating measured seizure-related parameters in the HA line as compared to acute PTZ administration. This phenomenon could be related to the compensatory changes in the cerebral expression of opioid peptides and their receptors in response to long-term blockage of the opioid system [42,55,56,57]. In addition, chronic PTZ administration would affect the expression of other receptors mediating excitatory and inhibitory neurotransmission; therefore, the effect of naloxone treatment could not be obvious. Long-term treatment of rats with PTZ had a significant impact on the densities of receptors for, e.g., GABA, glutamate and adenosine [58]. In other reports, kainic acid-evoked status epilepticus in the rat hippocampus affected GABA and glutamate receptor expression [59,60].

The next stage of our study involved selective blockage of particular opioid receptors using specific opioid antagonists administered before injection of PTZ. Selective µ opioid receptor blockade by β-funaltrexamine (β-FNA) shortened seizure latency but had a mitigating effect on seizure intensity in HA mice. However, these observations were only seen at the highest dose of PTZ. The anticonvulsive action of this compound was noted in another report [19]. Furthermore, Tortella and co-workers demonstrated that rats injected icv. with enkephalin for seizure induction, when pre-treated with β-FNA, exhibited a decrease in the proconvulsive effect of enkephalin detected by electroencephalography (EEG) [61].

Taking into account selective inhibition of κ and δ opioid receptors (with nor-binaltorphimine and naltrindole, respectively), their administration intensified convulsions, but prolonged seizure latency only in HA mice upon PTZ injection. In a mouse model of PTZ kindling [62], nor-binaltorphimine (5 mg/kg) administered sc. 24 h before PTZ (45 mg/kg) decreased seizure threshold and increased frequency of seizure onset within 30 min following PTZ, but only on the first day of 10-day kindling with PTZ (thus it may be treated as an acute model if considering only the time points of the onset of a significant effect of nor-binaltorphimine (nor-BNI)). Previous studies have shown that nor-BNI produced a selective and long-lasting (3 weeks) blockade of κ opioid receptors [63]. In our study, mice received ip. nor-BNI (20 mg/kg) 20 min before PTZ ip. In both our and a cited study, nor-BNI seemed to display rather proconvulsive properties. Seizure-promoting activity also displayed δ opioid receptor antagonist naltrindole (11.4 mg/kg) in HA mice after PTZ injection. The results obtained in another report [64], where naltrindole (0.3–10 mg/kg) decreased seizure threshold in a dose-dependent manner in the maximal electroshock test, support our observations. On the contrary, in another study [65], mice treated with low-dose naltrindole (1–2 mg/kg) demonstrated elevation of PTZ-evoked seizure thresholds by about 25% as compared to untreated animals. As the authors emphasized, they used different strains of mice in this study (i.e., C57BL/6J) as compared to their previous study (i.e., mice C57BL/6N) [16]. While C57BL/6J mice showed elevated seizure thresholds in response to naltrindole, C57BL/6N mice did not. Furthermore, the seizure threshold in C57BL/6J mice was unaffected following SNC80 treatment, while C57BL/6N mice showed a decrease in seizure threshold [65]. The authors claimed that these discrepancies related to the effect of δ receptor blockage might originate from strain-specific differences, also described in another report [66].

The next approach in our experimental sets was the activation of the endogenous opioid system in mice using several agonists, followed by the induction of seizures. We use morphine, which easily crosses the blood-brain barrier (8.8 mg/kg), biphalin (25 mg/kg) with impaired ability to penetrate into CNS and partial agonist buprenorphine (13.8 mg/kg).

Injection of a fixed dose of morphine followed by acute administration of PTZ at different doses had no effect on seizure intensity and latency in HA and LA mice. Consistently, in another report [67] neither a morphine dose of 7.5 mg/kg (close to the dose level in our experiment) nor a 2-fold greater amount affected seizure threshold. Interestingly, lower doses of morphine (0.5–3 mg/kg) or much greater concentrations (20–100 mg/kg) significantly increased or decreased seizure threshold, respectively, which implied a dose-dependent, biphasic modulation of clonic seizures upon morphine delivery [67]. On the contrary, chronic morphine treatment (8.8 mg/kg) with PTZ at different doses of HA and LA mice revealed a known biphasic response [68]. Long-term morphine administration with 50 mg/kg of PTZ significantly decreased seizure intensity in HA mice in comparison to HA mice chronically receiving only PTZ at the same dose. PTZ at 75 mg/kg in combination with morphine in chronic procedures increased seizure intensity in mice of both lines. Further PTZ dose escalation to 100 mg/kg, in combination with morphine, had no significant effect, likely related to the “ceiling effect”, a phenomenon widely described in the literature, e.g., regarding buprenorphine [69].

Administration of either buprenorphine (14 mg/kg) or biphalin followed by acute PTZ injection caused earlier seizure occurrence of higher intensity in HA mice than in their LA counterparts. In the nociceptive assays, buprenorphine showed a submaximal, or even bell-shaped dose-response curve, depending on the type and intensity of the stimulus, which is specific for partial agonists/antagonists [70]. In our study, both µ opioid receptor activation and blockage of κ and δ opioid receptors by a fixed dose of buprenorphine produced a proconvulsive effect; however, lower or higher doses of this compound might lead to completely different outcomes, taking into consideration the specific properties of buprenorphine. Further studies with different buprenorphine doses would reveal whether known drug activity in nociception exerts a biphasic mode of action in epilepsy. Considering biphalin, as a non-selective, high-affinity agonist for all three opioid receptor types [71], this peptide also displayed proconvulsive action. Our study is the first that describes biphalin’s properties in terms of seizure modulation, and there are no other reference studies that report biphalin’s effect in epilepsy. Since opioids, similarly to morphine, may prevent seizures but also promote them by their inhibitory influence on GABAergic interneurons and GABA release [19,72], the proconvulsive effect of biphalin observed by us is possible and explainable.

In studies modeling human disorders, including epilepsy, usage of the drug being applied in clinical practice is highly desirable to suggest its known effect on the results of tested compounds. In line with such an approach, we employed diazepam (DZP) as a reference substance. Pre-treatment of HA and LA mice with DZP significantly reduced PTZ-induced seizure intensity, particularly in LA mice with low endogenous opioid system activity. We suspect that the selection procedure might contribute to the development of compensatory, much better working inhibitory neurotransmission involving, e.g., the GABAergic system in LA mice in response to a disabled opioid system [19,72]. Another speculative hypothesis for this phenomenon could be a more profound benzodiazepine-induced release of anticonvulsive adenosine in conditions of opioid system inhibition, e.g., by naloxone. It would compensate for the missing modulatory effect of proper and undisturbed functioning of the opioid system [73]. Interestingly, opioid receptor blockade by naloxone potentiated the anticonvulsant activity of diazepam at 1 mg/kg in HA and LA mice to the level observed for animals administered with 5 mg/kg of diazepam. One possible mechanism involves a naloxone-dependent increase of adenosine in the synaptic cleft and activation of adenosine A1 receptors [73]. In the LA line, which is mostly insensitive to opioid receptor ligands, it is highly possible that naloxone’s effect originates from a non-opioid mechanism. In a study on rats, aiming to develop a therapy for tramadol overdose (manifested by seizures), the most effective remedy was a combination of naloxone with diazepam, as compared to the effects of separately administered drugs [74].

Our experiments confirmed that both inhibition and activation of the opioid system may induce seizures. As we documented in our study, the endogenous opioid system plays a significant modulatory role, mediated mainly by δ and κ opioid signaling.

## 4. Material and Methods

### 4.1. Animal Housing and Selection Procedure

All experiments were conducted on 8–10-week-old outbred male Swiss Webster mice selected for 87 generations towards high (HA) and low (LA) analgesia triggered by swim-induced stress. The animals were housed in the animal facility of the Institute of Genetics and Animal Biotechnology (Polish Academy of Sciences). Pups were weaned at 3 weeks of age and maintained in standard cages (5 littermates per cage) lined with fine sawdust bedding. Free access to tap water and pellet food Labofeed H (Morawski Co., Kcynia, Poland) was provided ad libitum. Cages rested on metal racks in a room with a constant temperature (22 ± 2 °C), air humidity (55 ± 5%) and artificial 12-h light/dark cycle (light phase starting at 7:00 am). The selection procedure was developed earlier by Panocka et al. [75] and relied on forcing the animals to swim for 3 min in a pool of water at a temperature of 20 °C. After the swim, mice were allowed to rest and dry for 3 min. in a cage lined with lignin. Next, their analgesia level (latency to thermal stimulus-response such as paw lifting) was measured after placing them on a 56 °C hot plate. The cut-off time was set to 60 s to avoid paw burns. Individuals with the shortest (up to 10 s) and the longest (50–60 s) paw withdrawal latencies were classified as the LA and HA lines, respectively. Experiments were performed upon approval from II Local Ethics Committee of Experiments on Animals (consent no. WAW2/58/2015) and in accordance with the International Association for the Study of Pain (IASP) guidelines.

### 4.2. Drugs

Morphine hydrochloride (Morph) was purchased from Polpharma (Warsaw, Poland); naloxone hydrochloride (NLX), β-funaltrexamine (β-FNA), naltrindole (NTI) and nor-binaltorphimine (nor-BNI) were delivered by TOCRIS (Bristol, United Kingdom); pentylenetetrazole (PTZ) and buprenorphine (Bupr) were obtained from Sigma-Aldrich (St. Louis, MO, USA). Biphalin (Biph) was synthesized and kindly delivered by Prof. A.W. Lipkowski from the Department of Neuropeptides, Mossakowski Medical Research Centre, Polish Academy of Sciences. Diazepam (DZP), sold on the market as Relanium, was purchased from Polfa (Warsaw, Poland).

### 4.3. Study Design

Pentylenetetrazole was used to induce seizures in HA and LA mice. To determine the role of the endogenous opioid system in seizure modulation, non-selective, as well as selective, opioid receptor antagonists were used prior to PTZ exposure. Naloxone hydrochloride was used as a non-selective opioid system blocker. The dose of naloxone (27.5 µmol/kg; that is, 10 mg/kg) was chosen based on our previous studies [76,77]. To selectively block µ, δ or κ opioid receptors, we used β-funaltrexamine, naltrindole or nor-binaltorphimine, respectively. To study the effect of opioid system activation on the proconvulsive effect of PTZ, mice were injected with morphine (preferential µ opioid receptor agonist), biphalin (potent non-selective opioid receptor agonist) and buprenorphine (partial µ opioid agonist and κ and δ opioid receptor antagonist). Both selective opioid receptor antagonists and agonists were administered ip. at doses equimolar to naloxone (27.5 µmol/kg). To study the interaction between the GABAergic and opioid systems in PTZ-induced seizure-like behavior, diazepam (a GABA_A_ agonist and a potent anticonvulsive medication) was given at 1 or 5 mg/kg alone or in combination with naloxone. All compounds were dissolved immediately before use in sterile saline solution and injected intraperitoneally (ip.) at a volume of 0.1 mL per 10 g of body weight.

### 4.4. Acute Treatment with PTZ

In the acute model of PTZ-induced seizures, mice received a single ip. injection of PTZ at 50, 75 and 100 mg/kg. Opioid receptor agonists/antagonists, DZP or saline were delivered ip. 20 min prior to PTZ administration. Immediately after PTZ injection, animals were observed for 30 min to assess seizure intensity and latency. The PTZ doses and experimental procedure were established based on our pilot study and previously reported protocols [78].

### 4.5. PTZ Kindling

The kindling model was introduced by daily PTZ injections (20 mg/kg, ip.) for 15 days. Each daily bolus of PTZ was preceded by an injection of either saline, non-selective opioid antagonist NLX, selective opioid antagonists or opioid agonists. On experimental day 16, the animals were treated with saline, NLX or morphine followed by PTZ at three different doses of either 50, 75 or 100 mg/kg to induce seizures. Then, animals were observed for 30 min following PTZ administration to determine the time of seizure onset and intensity.

### 4.6. Assessment of Seizure Intensity

Seizure intensity was determined according to Racine’s scale: stage 0—no response; stage 1—chewing or facial twitches; stage 2—chewing and head nodding or wet dog shakes; stage 3—unilateral forelimb clonus; stage 4—bilateral forelimb clonus and rearing; stage 5—bilateral forelimb clonus, rearing and falling. During the assessment, animals were maintained in individual cages in a quiet room.

### 4.7. Statistical Analysis

Data from all experiments were analyzed with STATISTICA 13.1 (StatSoft Inc., Tulsa, OK, USA). The normality of the collected data was tested by the Shapiro–Wilk test. Then, a one-, two- or three-way analysis of variance (ANOVA) was performed with the following independent factors: mouse line (HA or LA), treatment and dose of PTZ. Post hoc comparisons were performed using Bonferroni’s test. All results were presented as means ± standard error of the mean (SEM). The statistical significance of the results was set at *p* < 0.05.

## Figures and Tables

**Figure 1 ijms-25-06978-f001:**
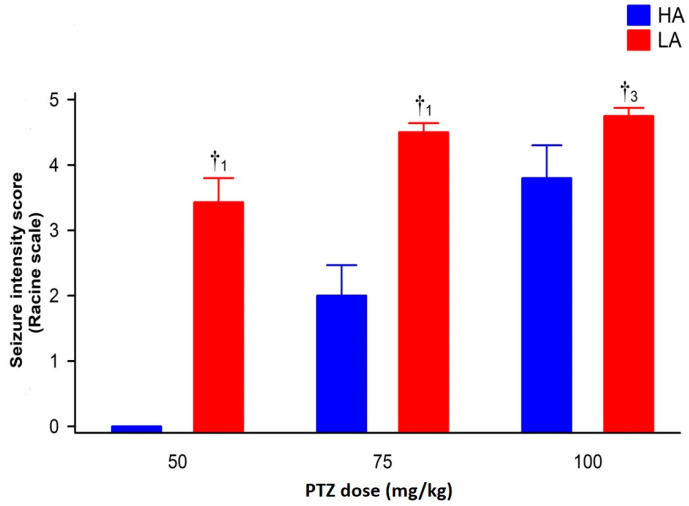
Seizure intensity in HA and LA mice assessed with Racine’s scale. Mice received ip. a single bolus of PTZ at 50, 75 and 100 mg/kg. Results were expressed as means ± SEM and analyzed with two-way ANOVA, followed by Bonferroni’s post hoc test. Results were considered significant at *p* < 0.05. The ‘†’ symbol represents comparisons between HA and LA mice for each dose of PTZ (n = 8–10). Lower-case numbers denote the significance level of †_1_
*p* < 0.001 and †_3_
*p* < 0.05.

**Figure 2 ijms-25-06978-f002:**
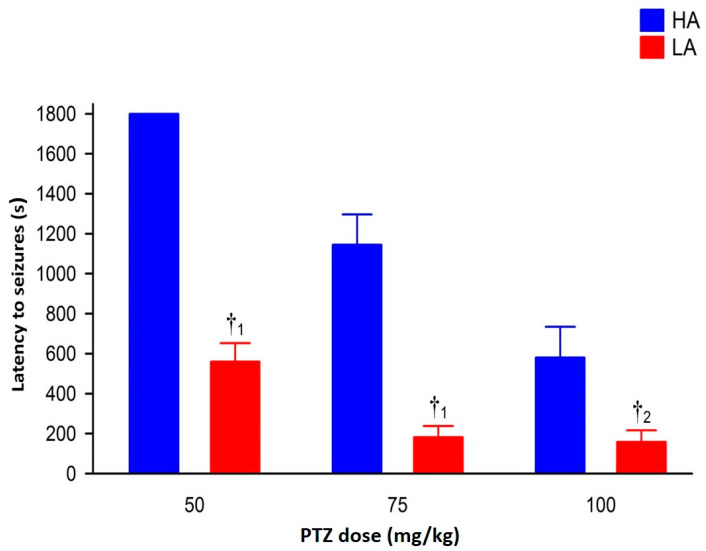
Seizure latency in HA and LA mice following a single bolus of PTZ ip. at 50, 75 and 100 mg/kg (n = 8–10). Results were expressed as mean seizure latency (s) ± SEM and analyzed with two-way ANOVA, followed by Bonferroni’s post hoc test. The statistical significance threshold was set at 0.05. Differences between HA and LA mice for each dose of PTZ were indicated with the ‘†’ symbol. Levels of significance were presented as lower-case numbers where: †_1_
*p* < 0.001 and †_2_
*p* < 0.01.

**Figure 3 ijms-25-06978-f003:**
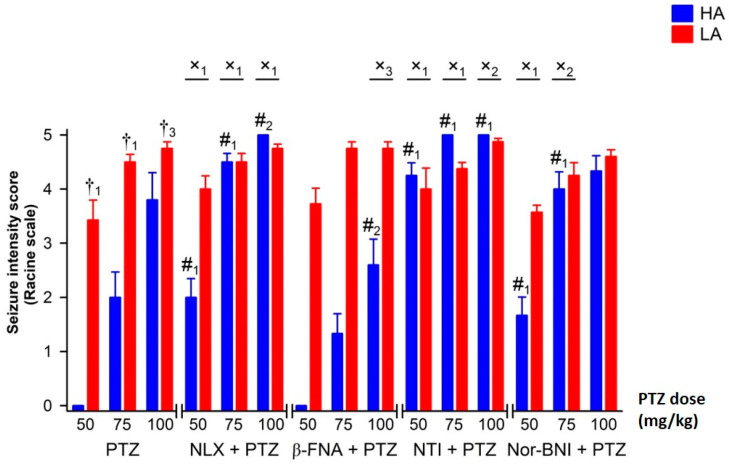
The effect of opioid receptor blockage on PTZ-induced seizure intensity in HA and LA mice according to Racine’s scale (n = 8–10). PTZ was delivered ip. as a single bolus at 50, 75 and 100 mg/kg. All opioid receptor antagonists, naloxone (NLX), β-funaltrexamine (β-FNA), naltrindole (NTI) and nor-binaltorphimine (nor-BNI), were administered ip. 20 min. prior to PTZ injection at an equimolar dose of 27.5 μmol/kg. Results were expressed as means ± SEM and analyzed with two- or three-way ANOVA, followed by Bonferroni’s post hoc test. The level of statistical significance was set at 0.05. Group comparisons were presented as follows: #_1_: *p* < 0.001; #_2_: *p* < 0.01 (HA/LA + PTZ vs. HA/LA + PTZ + antagonist); †_1_: *p* < 0.001; †_3_: *p* < 0.05 (HA + PTZ vs. LA + PTZ); x_1_: *p* < 0.001; x_2_: *p* < 0.01; x_3_: *p* < 0.05 (HA + PTZ vs. LA + PTZ and HA + PTZ + antagonist vs. LA + PTZ + antagonist).

**Figure 4 ijms-25-06978-f004:**
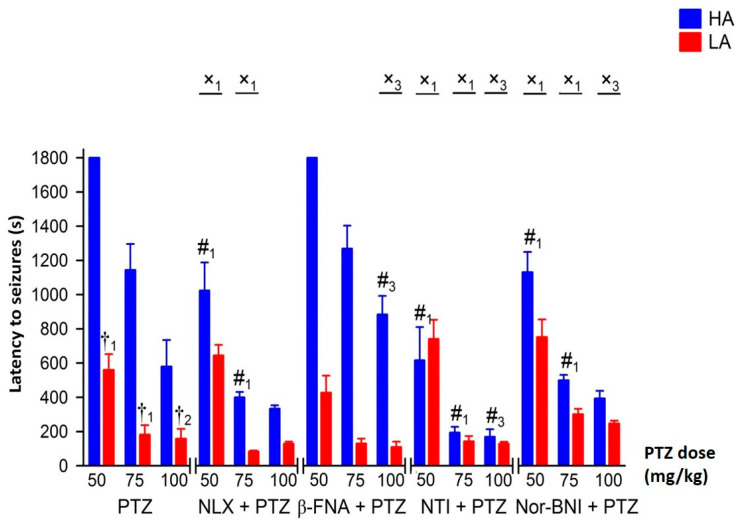
The effect of opioid receptor blockage on seizure latency in HA and LA mice following PTZ injection (n = 8–10). PTZ was delivered ip. as a single bolus at 50, 75 and 100 mg/kg. All opioid receptor antagonists, naloxone (NLX), β-funaltrexamine (β-FNA), naltrindole (NTI) and nor-binaltorphimine (nor-BNI), were administered for 20 min. prior to PTZ at an equimolar dose of 27.5 μmol/kg. Results were expressed as mean seizure latency (s) ± SEM and analyzed with two- or three-way ANOVA, followed by Bonferroni’s post hoc test. The level of statistical significance was set at 0.05. Group comparisons were presented as follows: #_1_: *p* < 0.001; #_3_: *p* < 0.05 (HA/LA + PTZ vs. HA/LA + PTZ + antagonist); †_1_: *p* < 0.001; †_2_: *p* < 0.01 (HA + PTZ vs. LA + PTZ); x_1_: *p* < 0.001; x_3_: *p* < 0.05 (HA + PTZ vs. LA + PTZ and HA + PTZ + antagonist vs. LA + PTZ + antagonist).

**Figure 5 ijms-25-06978-f005:**
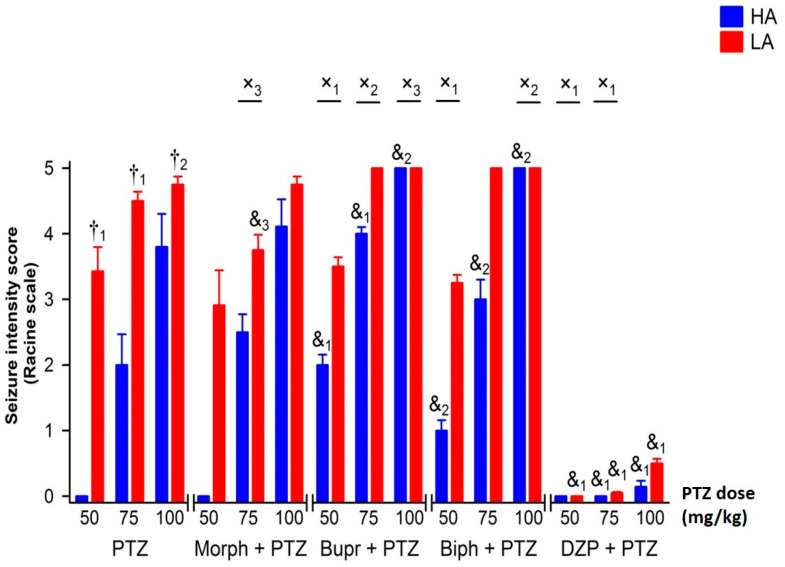
The effect of opioid receptor agonists on the intensity of PTZ-induced seizures in HA and LA mice (n = 8–10). Seizure intensity was evaluated by means of the Racine scale. PTZ was administered ip. to HA or LA mice as a single bolus at 50, 75 or 100 mg/kg. Morphine (Morf), buprenorphine (Bupr) and biphalin (Biph) were delivered 30 min before PTZ at an equimolar dose of 27.5 μmol/kg. Results were expressed as means ± SEM and analyzed with two- or three-way ANOVA, followed by Bonferroni’s post hoc test. The level of statistical significance was set at 0.05. Group comparisons were presented as follows: &_1_: *p* < 0.001; &_2_: *p* < 0.01; &_3_ (HA/LA + PTZ vs. HA/LA + PTZ + agonist); †_1_: *p* < 0.001; †_2_: *p* < 0.01 (HA + PTZ vs. LA + PTZ); x_1_: *p* < 0.001; x_2_: *p* < 0.01; x_3_: *p* < 0.05 (HA + PTZ vs. LA + PTZ and HA + PTZ + agonist vs. LA + PTZ + agonist).

**Figure 6 ijms-25-06978-f006:**
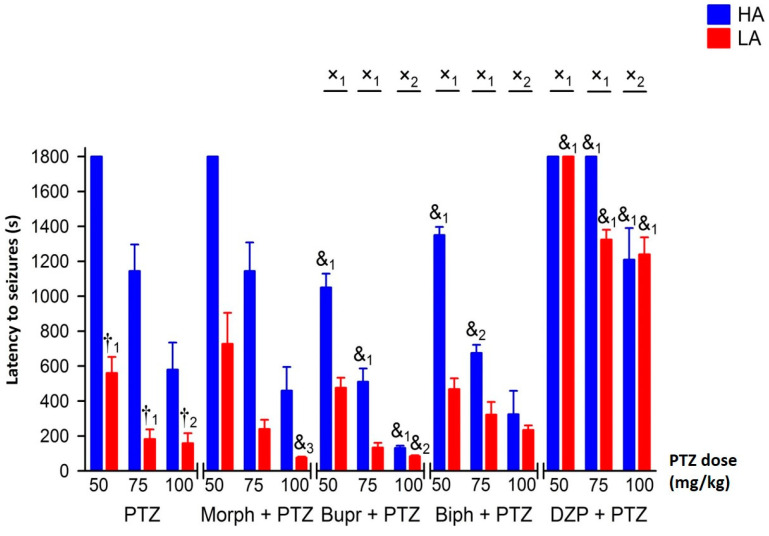
The effect of opioid receptor agonists on seizure latency in HA and LA mice following PTZ injection (n = 8–10). PTZ was administered ip. as a single bolus at 50, 75 and 100 mg/kg. Morphine (Morf), buprenorphine (Bupr) and biphalil (Biph) were injected ip. 30 min before PTZ at an equimolar dose of 27.5 μmol/kg. Results were expressed as mean seizure latency (s) ± SEM and analyzed with two- or three-way ANOVA, followed by Bonferroni’s post hoc test. The level of statistical significance was set to 0.05. Group comparisons were presented as follows: &_1_: *p* < 0.001; &_2_: *p* < 0.01; &_3_ (HA/LA + PTZ vs. HA/LA + PTZ + agonist); †_1_: *p* < 0.001; †_2_: *p* < 0.01 (HA + PTZ vs. LA + PTZ); x_1_: *p* < 0.001; x_2_: *p* < 0.01 (HA + PTZ vs. LA + PTZ and HA + PTZ + agonist vs. LA + PTZ + agonist).

**Figure 7 ijms-25-06978-f007:**
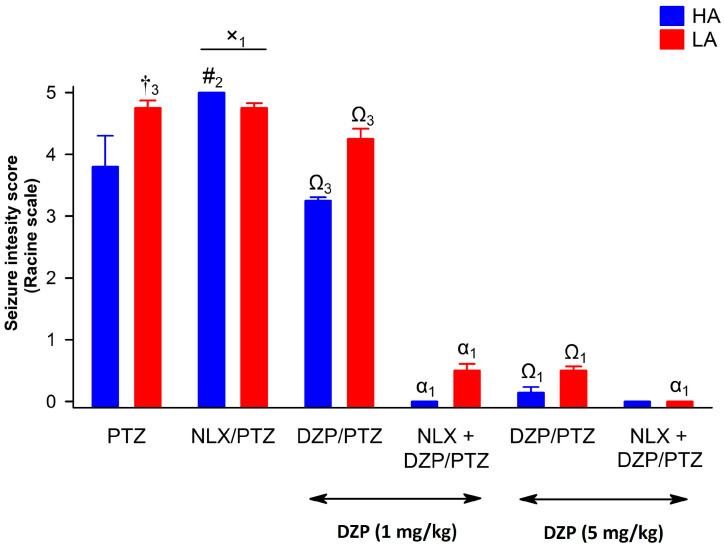
The effect of naloxone (NLX) and diazepam (DZP) on seizure intensity in HA and LA mice injected ip. with a single dose of PTZ at 100 mg/kg (n = 8–10). Seizure intensity was evaluated by means of the Racine scale. Diazepam was administered ip. at a dose of 1 or 5 mg/kg 15 min after NLX injection (27.5 μmol/kg), but 15 min prior to PTZ. Results were expressed as means ± SEM and analyzed with two- or three-way ANOVA, followed by Bonferroni’s post hoc test. The level of statistical significance was set to 0.05. Group comparisons were presented as follows: #_2_: *p* < 0.01 (HA/LA + PTZ vs. HA/LA + PTZ + NLX); †_3_: *p* < 0.05; x_1_: *p* < 0.05 (HA + PTZ + NLX vs. LA + PTZ + NLX); Ω_1_
*p* < 0.001; Ω_3_
*p* < 0.05 (HA/LA + PTZ vs. HA/LA + PTZ + DZP); α_1_ (HA/LA + PTZ + DZP vs. HA/LA + PTZ + DZP + NLX).

**Figure 8 ijms-25-06978-f008:**
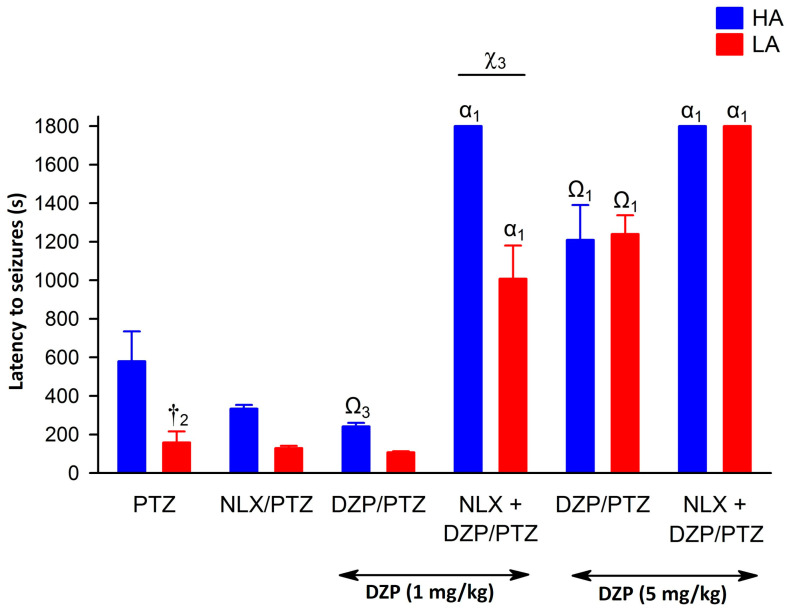
The effect of naloxone (NLX) and diazepam (DZP) on seizure latency in HA and LA mice injected ip. with a single dose of PTZ at 100 mg/kg (n = 8–10). Seizure threshold was expressed as mean seizure latency [s] ± SEM. Diazepam (1 and 5 mg/kg) was administered ip. at 15 min after NLX injection (27.5 μmol/kg), but 15 min prior to PTZ. Results were expressed as mean seizure latency (s) ± SEM analyzed with two- or three-way ANOVA, followed by Bonferroni’s post hoc test. The level of statistical significance was set to 0.05. Group comparisons were presented as follows: †_2_: *p* < 0.01 (HA vs. LA for PTZ-treated groups); Ω_1_
*p* < 0.001; Ω_3_
*p* < 0.05 (HA/LA + PTZ vs. HA/LA + PTZ + DZP); α_1_ (HA/LA + PTZ + DZP vs. HA/LA + PTZ + DZP + NLX). χ_3_ indicates the effect of NLX on the anticonvulsant potential of DZP between individuals of the HA and LA lines (line × DZP × NTX interaction effect); *p* < 0.001.

## Data Availability

The original data obtained in the study are included in this paper and Appendix A. Any other data may be available upon request upon e-mail to the corresponding author of this article.

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
