# Peer review of "Susceptibility to Pentylenetetrazole-Induced Seizures in Mice with Distinct Activity of the Endogenous Opioid System"

_ijms, 2024, doi:10.3390/ijms25136978_

Round 1
Reviewer 1 Report
Comments and Suggestions for Authors
The authors of the manuscript "Susceptibility to pentylenetetrazole-induced seizures in mice with distinct activity of endogenous opioid system" describe the role of opioid tone in PTZ-induced seizures.
The hypothesis is clearly defined and the manuscript is well written with a minimum of typos, e.g. line 74 (typo in reference), line 287 (...alone. , The effect ...). The introduction is extensive. The project itself would require many more experiments, which cannot be described in a manuscript and are far beyond the scope of this manuscript. The experiments are well proposed, the results are robust, and I have not identified any significant issues for revision.
However, I do have a few technical comments for consideration.
Comment 1 - Line 169: Please add a comment or reference regarding the dose of naloxone. For example: the dose was chosen based on ...
Comment 2 - line 180-182: According to your description, the drugs were dissolved in 0.1 ml per 10 g body weight. In other words, the drug concentration is different in each sample. This is just a technical comment that the absorption of the drug may be different depending on the concentration used. It may not affect your study but should be considered for a combination of compounds.
Comment 3 - When proposing the role of opioid signalling in PTZ-induced seizures, the authors should consider including a paragraph on approved antiepileptic drugs and their known effects on opioid signalling.
Author Response
Dear Reviewer,
Thank you for all your valuable remarks. Please, see below our responses to your questions/concerns.
Comment 1: Line 169: Please add a comment or reference regarding the dose of naloxone. For example: the dose was chosen based on ...
Response 1: Thank you for valuable comment. The dose of naloxone hydrochloride (27.5 uM/kg that is 10 mg/kg) was chosen based on our previous studies:
- Nawrocka A, Poznański P, Łazarczyk M, Gorzałczyński M, Skiba D, Wolińska R, Bujalska-Zadrożny M, Lutfy K, Sadowski B, Sacharczuk M. The Influence of Cross-Fostering on Alcohol Consumption and Depressive-Like Behaviors in HA and LA Mice: The Role of the Endogenous Opioid System. Brain Sci. 2021 May; 11(5): 622. Published online 2021 May 13. doi: 10.3390/brainsci11050622.
- Poznanski P, Lesniak A, Korostynski M, Szklarczyk K, Łazarczyk M, Religa P, Bujalska-Zadrozny M, Sadowski B, Sacharczuk M. Delta-opioid receptor antagonism leads to excessive ethanol consumption in mice with enhanced activity of the endogenous opioid system. Neuropharmacology. 2017 May 15:118:90-101. doi: 10.1016/j.neuropharm.2017.03.016. Epub 2017 Mar 18.
Please see the line 461-463 of the revised manuscript, where the missing information has been included.
Comment 2: line 180-182: According to your description, the drugs were dissolved in 0.1 ml per 10 g body weight. In other words, the drug concentration is different in each sample. This is just a technical comment that the absorption of the drug may be different depending on the concentration used. It may not affect your study but should be considered for a combination of compounds.
Response 2: Thank you for this valuable suggestion. We share this point of view with the Reviewer and agree completely with the notion. Based on our experience with the same compounds and their similar concentrations used by us previously in other studies involving the same animal model, the time that elapses from the moment of drug injection to the planned tests execution seems to be fixed accurately and allows for sufficient absorption of the drugs to uncover and record the drugs’ effects.
Comment 3: When proposing the role of opioid signalling in PTZ-induced seizures, the authors should consider including a paragraph on approved antiepileptic drugs and their known effects on opioid signalling.
Response 3: Thank you for your valuable comment. International League Against Epilepsy website refers to and recommends multiple guidelines for therapy of epilepsy, including NICE guidelines (NG217) published on 27 April 2022. For the first line monotherapy of generalized tonic-clonic seizures, sodium valproate, lamotrigine or levetiracetam are offered. Add-on first line treatment, if monotherapy is unsuccessful, incudes clobazam, lamotrigine, levetiracetam, perampanel, sodium valproate and topiramate. Furthermore, second line add-on treatment options are brivaracetam, lacosamide, phenobarbital, primidone, zonisamide. Other antiseizure medications that are well-known and relatively historical, but may exacerbate seizures in people with absence, or myoclonic seizures are: carbamazepine, gabapentin, lamotrigine (for myoclonic seizures), oxcarbazepine, phenytoin, pregabalin, tiagabine, vigabatrin. In case of focal seizures with or without evolution to bilateral tonic-clonic seizures, for monotherapy lamotrigine or levetiracetam are recommended. If ineffective, following drugs shall be considered: carbamazepine, oxcarbazepine, zonisamide. If second-line monotherapies are unsuccessful in patients with focal seizures, lacosamide as third-line monotherapy is recommended. If monotherapy is unsuccessful in patients with focal seizures, one of the following first line add-on treatment options shall be considered: carbamazepine, lacosamide, lamotrigine, levetiracetam, oxcarbazepine, topiramate, zonisamide. As we see, many medications are recommended in various clinical cases, therefore for the need of our manuscript, we selected most commonly recommended drugs for first, second or third line therapy of epilepsy adopted commonly in present clinical practice, namely: sodium valproate, lamotrigine, levertiracetam, and some relatively “old” medications that have been widely tested for many years both in clinical trials and animal studies as: phenytoin, carbamazepine, gabapentin, pregabalin and clonazepam.
Please refer to the separate subsection of the introduction section (line 100-148) for checking requested paragraph.
Thank you for spending your time and contributing to the quality improvement of our manuscript.
Reviewer 2 Report
Comments and Suggestions for Authors
This manuscript addresses a critical issue in epilepsy research, investigating the role of the endogenous opioid system in seizure susceptibility using two mouse lines selected for low (LA) and high (HA) opioid system activity. The study uses pentylenetetrazole (PTZ) to induce seizures and explores the impact of naloxone and diazepam (DZP) on seizure control. The findings suggest that a well-regulated opioid system is crucial for preventing generalized tonic-clonic seizures and that combining naloxone with DZP could offer a DZP-sparing effect, which is important for reducing benzodiazepine addiction liability. The manuscript presents a well-conducted study with significant findings that contribute to the understanding of the endogenous opioid system's role in seizure control. I only have two minor comments:
1. The discussion section is too long and verbose. Some of the detailed explanations and contextual information currently in the discussion could be integrated into the results section where they directly relate to the data presented. This will make the discussion more focused and impactful.
2. There are too many figures in the manuscript, which can overwhelm the reader and dilute the impact of the key findings. I recommend using large figures with sub-plots to present the data more effectively.
Author Response
Dear Reviewer,
Please, see below our responses to your suggestions. Thank you for spending your time and contributing to the improvement of our manuscript.
Comment 1: The discussion section is too long and verbose. Some of the detailed explanations and contextual information currently in the discussion could be integrated into the results section where they directly relate to the data presented. This will make the discussion more focused and impactful.
Response 1: Thank you for this valuable suggestion. We have revised the whole discussion according to your remarks. Please check corrected discussion section in the revised version of the manuscript (line 291-429).
Comment 2: There are too many figures in the manuscript, which can overwhelm the reader and dilute the impact of the key findings. I recommend using large figures with sub-plots to present the data more effectively.
Response 2: To reduce number of figures and make the manuscript more legible and clear, we have shifted Figures (previously numbered as 9, 10, 11, 12) to the Supplementary Materials. Thank you for this valuable advice.